# 5-Oxo-ETE/OXER1: A Link between Tumor Cells and Macrophages Leading to Regulation of Migration

**DOI:** 10.3390/molecules29010224

**Published:** 2023-12-31

**Authors:** Konstantina Kalyvianaki, Evangelia Maria Salampasi, Elias N. Katsoulieris, Eleni Boukla, Amalia P. Vogiatzoglou, George Notas, Elias Castanas, Marilena Kampa

**Affiliations:** Laboratory of Experimental Endocrinology, School of Medicine, University of Crete, 71500 Heraklion, Greece; kalyvianakikon@gmail.com (K.K.); evasalampasi@gmail.com (E.M.S.); ekatsoulieris@gmail.com (E.N.K.); dimelenh@hotmail.com (E.B.); amaliavogiatzoglou@gmail.com (A.P.V.); gnotas@uoc.gr (G.N.)

**Keywords:** OXER1, androgens, 5-oxo-ETE, migration, macrophages, prostate cancer, TME

## Abstract

Chronic inflammation is an important factor in the development of cancer. Macrophages found in tumors, known as tumor associated macrophages (TAMs), are key players in this process, promoting tumor growth through humoral and cellular mechanisms. 5-oxo-6,8,11,14-eicosatetraenoic acid (5-oxo-ETE), an arachidonic acid metabolite, has been described to possess a potent chemoattractant activity for human white blood cells (WBCs). The biological actions of 5-oxo-ETE are mediated through the GPCR 5-oxo-6E,8Z,11Z,14Z-eicosatetraenoic acid receptor (OXER1). In addition, we have previously reported OXER1 as one of the membrane androgen receptors with testosterone antagonizing 5-oxo-ETE’s actions. OXER1 is highly expressed in inflammatory cells and many normal and cancer tissues and cells, including prostate and breast cancer, promoting cancer cell survival. In the present study we investigate the expression and role of OXER1 in WBCs, THP-1 monocytes, and THP-1 derived macrophages, as well as its possible role in the interaction between macrophages and cancer cells (DU-145 and T47D). We report that OXER1 is differentially expressed between WBCs and macrophages and that receptor expression is modified by LPS treatment. Our results show that testosterone and 5-oxo-ETE can act in an antagonistic way affecting Ca^2+^ movements, migration, and cytokines’ expression in immune-related cells, in a differentiation-dependent manner. Finally, we report that 5-oxo-ETE, through OXER1, can attract macrophages to the tumor site while tumor cells’ OXER1 activation in DU-145 prostate and T47D breast cancer cells, by macrophages, induces actin cytoskeletal changes and increases their migration.

## 1. Introduction

The interplay between the tumor mass and the surrounding tissue, called the tumor microenvironment (TME), regulates the fate of tumor cells. TME consists of a large array of cell types, embedded in an amorphous collagen matrix, including fibroblasts, endothelial cells, and many immune-related cells. Lymphocytes, NK-cells, neutrophils, and macrophages exist in this environment and regulate tumor growth/proliferation or death (reviewed in [1]). Among these immune-related cells, tumor-associated macrophages (TAM) constitute about 30–50% of immune cells and are directly related to the evolution of solid tumors [2].

The understanding of the origin of TAMs has evolved over the years. Most TAMs were suggested to derive from circulating monocytes [3]. However, recently, a primary role of resident tumor macrophages (RTMs) has been acknowledged [4], presenting a differential location within the TME, in different cancers [1]. In a recent work, using single-cell analysis, Cao and collaborators confirmed that TAMs in ovarian tumors originate both from RTMs and infiltrating monocytes [5]. The same authors also reported a high proliferation rate of the macrophage population, expressing a continuum of markers between M1 and M2 phenotypes.

The role of TAMs has been extensively studied in recent years, especially in view of establishing novel, targeted therapies (reviewed in [2,6]). Major interventions targeting TAMs address their migration, either affecting blood monocytes or RTMs within the tumor [7,8,9,10].

A major chemoattractant identified in recent years is 5-oxo-6E,8Z,11Z,14Z-eicosatetraenoic acid (5-oxo-ETE). 5-oxo-ETE is a product of arachidonic acid metabolism by 5-lipoxygenase (ALOX5) and the not-yet-cloned 5-hydroxyeicosanoid dehydrogenase (5-HEDH). Its biological actions are mediated through the GPCR 5-oxo-ETE receptor (OXER1) [11,12,13,14,15,16]. Both 5-oxo-ETE and OXER1 can be found in human white blood cells (lymphocytes, monocytes, neutrophils, eosinophils) and various tissues [17,18,19,20,21], where 5-oxo-ETE acts as a potent chemoattractant [22,23,24]. Interestingly, OXER1 is further implicated in osmotic-mediated wound sensing [25], while redox activation of ALOX5 can lead to long-lasting production of 5-oxoETE, which directly attracts distant leukocytes to the site of injury via OXER1 signaling, in zebrafish embryos [26]. Finally, we have shown that OXER1 is up-regulated by its ligand 5-oxo-ETE at the leading edge of the wound in human cancer epithelial cells (prostate, breast, and hepatocellular), mimicking the activation/migration phase of healing [12].

In addition to 5-oxo-ETE, OXER1 also mediates membrane-initiated effects of androgens [11,15,27]. Testosterone binding to OXER1 occurs in the same binding groove as 5-oxo-ETE [11,28,29], with testosterone inhibiting the G_αi_-induced cAMP inhibition [11,28,29], while 5-oxo-ETE inhibits testosterone-induced Ca^2+^ release from intracellular stores [15]. Interestingly, 5-oxo-ETE enhances cell migration [11,12] by modifying the actin cytoskeleton [11], a G_βγ_-mediated action [29]. This effect is completely reversed by testosterone [11,29]. It is to be noted that all these actions of testosterone are independent of the steroid action on classical androgen receptors.

In the present work, we investigated the role of 5-oxo-ETE through OXER1 in the migration of human macrophages. We report that OXER1 plays a significant role in M1 and M2 migration, while an interplay of epithelial cancer cells and macrophages through this receptor may be implicated in the migratory/metastatic capacity of tumors.

## 2. Results

### 2.1. Soluble Factors Secreted by Cancer Cells Attract M1 and M2 Macrophages

Differentiated M1 and M2 THP-1 cells were incubated in co-culture with DU-145 prostate cancer cells in transwell cell culture inserts. The migration of both macrophage cell types was significantly enhanced (Figure 1), compared with macrophages cultured in the absence of DU-145 cells, indicating that humoral factors secreted by cancer cells mediate the chemoattraction of macrophages. The implication of OXER1 was confirmed by synchronous incubation with Gue 1654 (a specific OXER1-G_βγ_ antagonist [30]) and testosterone-BSA (a non-permeable testosterone analog), which inhibited the phenomenon.

### 2.2. Expression of OXER1 by Macrophages during Their Differentiation

In view of the above results, we investigated the presence of OXER1 in circulating human monocytes and in THP-1 cells during their differentiation. As shown in Figure 2, human circulating monocytes express OXER1. Furthermore, we confirmed the presence of OXER1 in lymphocytes and neutrophils, as previously reported [23,31], with the latter expressing the highest levels (Figure 2). Interestingly, OXER1 expression in lymphocytes was lower than neutrophils but was not as low as previously reported, possibly due to interindividual variations. Treatment with lipopolysaccharide (LPS), a pro-inflammatory component of the Gram-negative bacteria cell wall, increased OXER1 expression (Figure 2B,C). It is known that LPS can induce the production of inflammatory cytokines and chemokines in different cell types, resulting in an acute inflammatory response [32,33]. Between the different leukocyte populations, increased OXER1 expression by LPS was more pronounced in monocytes. Furthermore, when THP-1, a monocytic leukemia cell line, was treated with different doses of LPS, OXER1 expression was equally increased in a dose-dependent manner (Figure 2E,F).

THP-1 cells were differentiated into M1 and M2 subtypes, and the expression of OXER1 was examined at both mRNA and protein level (Figure 3A,B). Different levels of OXER1 mRNA expression were detected in all tested cell types. Interestingly THP-1 cells expressed high levels of OXER1. During differentiation, M0 cells exhibited very low levels of the receptor, while higher OXER1 levels were found in differentiated M1 and M2 cells. We note that M0, M1, and M2 cells exhibited cytoplasmic as well as membrane expression of the receptor (Figure 3C).

### 2.3. Effect of OXER1 Activation on THP1 Differentiation

Monocytes can be polarized into different subsets of macrophages, according totheir response to inflammatory stimuli. Traditionally, pro-inflammatory stimuli promote differentiation into macrophages with the pro-inflammatory M1-like phenotype, while anti-inflammatory stimuli lead to macrophages with the M2-like phenotype. As reviewed by Chistiakov’s team [34], IFN-γ leads to M1 macrophages with the most potent pro-inflammatory properties and IL-4 results in the most typical M2-like phenotype. To investigate a possible role of OXER1 in THP-1 monocyte differentiation, PMA-treated cells were incubated with 5-oxo-ETE or testosterone –BSA 6 h before the addition of IFNγ and LPS or IL-4. We noted that no significant differences could be detected. Morphologically, no changes were observed and the main markers of differentiation used for macrophage phenotype identification, notably CD68 for M1 and CD206, were not altered by OXER1 ligands (Appendix A), suggesting that OXER1 is not a differentiation signal in these cells.

### 2.4. Effect of OXER1 Activation on the Expression of Humoral Mediators

M1 and M2 macrophages were incubated with OXER1 agonist (5-oxo-ETE) and antagonists (testosterone–BSA and Gue 1654, a specific G_βγ_ OXER1 antagonist, [30]), and the synthesis (through mRNA expression) of different cytokines was assayed. The expression levels of TNF-α, IL-6, and IL-1b were examined for M1 cells and TGF-β, Arg1, and IL-10 for M2 cells. Three-hour treatment with 5-oxo-ETE, testo–BSA, or Gue 1654 did not alter mRNA levels of the previously mentioned cytokines, except those of TGF-β (Appendix A). 5-oxo-ETE increased TGF-β expression, with both testosterone–BSA and Gue 1654 inhibiting the effect. In contrast, when THP-1 cells were treated with testosterone–BSA, the expression of cytokines IL-1b, TNF-α, IL-10, and IL-12 was increased (Appendix A). This effect was inhibited when the cells were simultaneously treated with 5-oxo-ETE and testosterone–BSA. These data suggest that OXER1 activation could possibly have a humoral effect in undifferentiated monocytes, which is not found after their differentiation into macrophage cells.

### 2.5. Differentiated Macrophages Express a Functional OXER1 Receptor

The absence of a humoral effect in differentiated macrophages questions the functionality of OXER1 expressed in these cells (Figure 2). We therefore investigated previously reported and validated OXER1 actions in other cell systems [11,15]. 5-oxo-ETE has been shown to alter intracellular calcium levels in human neutrophils [23] and cancer cells [15].

#### 2.5.1. Ca^2+^ Release from Intracellular Sources

Intracellular calcium changes in THP-1 cells and THP-1-derived macrophages were recorded using fluo-4 AM in a calcium-free environment, to record Ca^2+^ release only from intracellular sources. Testosterone–BSA increased intracellular calcium in M1 and M2 cells, an effect reversed by adding 5-oxo-ETE (Figure 4). In contrast, 5-oxo-ETE alone had no effect, in accordance with our previous results in prostate cancer cells [15], even though in neutrophils we [15] and others [23] have shown 5-oxo-ETE to increase calcium levels. This difference could be attributed to different signaling machinery of the different cell types. Testosterone–BSA-induced calcium changes were also partially inhibited by Gue 1654. We note that DHT did not increase calcium levels in an aspect similar to testosterone–BSA and cyproterone acetate, an AR antagonist, did not block testosterone–BSA action. This further indicates that the effect of testosterone–BSA is mediated solely by a membrane androgen receptor.

#### 2.5.2. G_αi_ Mediated cAMP Modifications

One of the main molecular events reported for OXER1 is the inhibition of cAMP generation, due to its coupling to a G_αi_ protein [19]. Surprisingly, in macrophages, this inhibitory effect was not observed, since 5-oxo-ETE did not abolish the forskolin-stimulated cAMP production, neither in M1 nor in M2 cells. Similarly, incubation with testosterone–BSA or simultaneous incubation with the two effectors did not affect cAMP-enhanced production by forskolin (Appendix A). This indicates an alternative mode of OXER1 action that possibly does not involve G_αi_/cAMP.

### 2.6. OXER1-Related Migration of Macrophages

Previous reports have shown that 5-oxo-ETE can rapidly modify actin polymerization on white blood cells, leading to chemotaxis and cell migration [35,36]. Actin polymerization and formation of protrusions known as lamellipodia and spike-like filopodia are also a response to migration stimulus in macrophages [9]. We have previously reported that 5-oxo-ETE triggers actin cytoskeleton rearrangements in prostate DU145 cancer cells, which highly express OXER1 but lack AR, leading to an increase in formation of stress fibers and an increased migratory capacity. This effect was reversed by the addition of testosterone–BSA [11]. Treatment of M1 (Figure 5) and M2 cells (Figure 6) with 5-oxo-ETE led to increased stress-fiber formation and lamellipodia regulation, with both testo–BSA and Gue 1654 reversing these effects.

Based on the finding that 5-oxo-ETE modified the actin cytoskeleton in M1 and M2 macrophages, its effect on M1 and M2 migratory capacity was investigated. We report that 5-oxo-ETE significantly enhanced the migration of both M1 (Figure 7A) and M2 (Figure 7C). Testosterone–BSA or Gue 1654 delayed the migratory rate compared with 5-oxo-ETE treatment. The latter’s effect was reversed when the cells were incubated with testosterone–BSA or Gue 1654 and 5-oxo-ETE (Figure 7B,D).

### 2.7. Reciprocal Attraction of Epithelial Cancer Cells by Macrophages, through 5-oxo-ETE/OXER1

We have previously reported that OXER1 plays a direct role in the process of cell migration and wound healing [11,12], with 5-oxo-ETE leading to an increased migratory capacity of prostate and breast cancer cells (DU-145, T47D), and testosterone–BSA and Gue 1654 inhibiting this action, an effect verified here (Appendix A). In addition, as presented above, OXER1 plays a role in macrophage migration (Figure 7). Taking into account these findings and that it has been previously reported that macrophages produce 5-LOX products, including 5-HETE, the precursor of 5-oxo-ETE [37], the involvement of 5-oxo-ETE/OXER1 on the migration of cancer cells in the presence of macrophages was also explored.

Treatment of DU-145 and T47D cancer cells with conditioned medium (CM) derived from M1 and M2 resulted in significant actin cytoskeleton re-organization in DU145 (Figure 8A) and T47D cells (Appendix A). Specifically, CM from macrophages affected actin organization and led to increased filopodia formation. Treatment with CM from macrophages supplemented with OXER1 antagonists, such as testo–BSA and Gue 1654, inhibited the initial actin cytoskeleton rearrangement, resulting in a return to the basal formation of the actin cytoskeleton. This indicates that secretory factors acting via OXER1 modulate the actin cytoskeleton. Interestingly, we report that the inverse phenomenon equally occurs: CM from either M1 or M2 macrophages enhanced the migratory capacity of prostate (Figure 8B) and breast cancer cells (Appendix A), as reported previously [38]. Incubation of cancer cells with CM from macrophages treated with zileuton, an inhibitor of ALOX5, a rate-limiting enzyme of 5-oxo-ETE biosynthesis, resulted in a noticeable reduction in the previously enhanced migration. Similarly, when Gue 1654 or testosterone–BSA were added during the treatment of DU-145 and T47D cells with CM from M1 or M2, the initial effect of the CM was reduced, confirming the implication of 5-oxo-ETE/OXER1 in this action. However, it cannot be excluded that another ALOX5 metabolite is also equally involved in stimulating 5-oxo-ETE synthesis by tumor cells, resulting in OXER1-mediated enhanced migration.

## 3. Discussion

Intrinsic effects of tumor cells are not the only factors regulating tumor growth, development, and metastasis. Other players forming the tumor microenvironment may influence oncogenesis and cancer progression. TME includes blood and lymphatic vessels, extracellular matrix, and distinct host cells, including fibroblasts and immune cells, in an intricate formation, surrounding and interacting with the tumor [4,39,40]. Among the immune cellular components of the TME, macrophages represent 30–50% of immune cells [2]. Their intrinsic properties, including phagocytosis, antigen presentation, and the secretion of an extensive array of immunomodulatory molecules, together with their differentiation and trans-differentiation potential has positioned them as a primary target for tumor immune manipulation and as a potential biomarker for tumor evolution, progression, and metastasis [1,4].

An intrinsic property of macrophages (tissue-resident or deriving from circulating monocytes [3,4,5]) is their capacity to migrate in the TME toward solid tumors. Several effector molecules have been proposed to act as factors promoting macrophage migration. Among them, 5-oxo-ETE and its receptor OXER1 have recently emerged as a major chemoattracting system. OXER1 is expressed in human leucocytes (lymphocytes, monocytes, neutrophils, and eosinophils) and various tissues [17,18,19,20,21], while a primary role of 5-oxo-ETE in asthma has been found, with specific OXER1 antagonists currently investigated as a potential novel treatment [41,42].

Here, we confirm the expression of OXER1 in different leucocyte populations, including circulating monocytes, in which its expression is most enhanced by the pro-inflammatory stimulus LPS, in a dose-dependent manner. In addition, during THP-1 cell differentiation toward M1 and M2 macrophages, the expression of OXER1 is the highest in naïve cells, decreasing during the M0 differentiation and further enhanced in both M1 and M2 populations, a result also found in a micro-array analysis of THP-1 cells (GSE67264 dataset, [43]). However, we could not identify a possible role of OXER1 in the differentiation process of THP-1 cells, although OXER1 in macrophages is active, as shown by the modulation of Ca^2+^ release from intracellular stores [15].

Cell migration depends heavily on altered polymerization and cellular redistribution of actin [44,45]. Here, we show that conditioned medium from DU-145 (prostate) and T47D (breast) epithelial cancer cells altered macrophage actin distribution and induced the migration of both M1 and M2 macrophages. The role of OXER1 in this phenomenon was documented by the direct role of 5-oxo-ETE and its reversion by adding the specific OXER1 antagonists Gue 1654 and testosterone–BSA. Interestingly, the role of OXER1/5-oxo-ETE is two-way, as tumor cells’ OXER1 activation by macrophages in DU-145 prostate and T47D breast cancer cells, on which OXER1 has been previously identified [21], induces actin cytoskeletal changes and migration. These findings are in accordance with what we and others have reported for the importance of OXER1 in prostate and breast cancer cell migration [12,27].

Cancer is considered a chronic inflammatory disease, and tumor-promoting inflammation is one of the hallmarks of cancer [46,47]. Here, we show that OXER1 is a prominent factor of inflammation, fulfilling one of the main hallmarks of inflammation [48], namely cell adhesion/migration. However, the exact role of each macrophage subtype (M1, M2) in cancer is far from completely elucidated. Both phenotypes have been implicated in cancer development, progression, and metastasis. Traditionally, M1 macrophages are known to possess primarily anti-tumor activities due to their ability to produce reactive nitrogen and oxygen intermediates and pro-inflammatory cytokines; M2 macrophages are considered pro-tumor, since they secrete anti-inflammatory cytokines that promote tumor progression by downregulating the immune response and stimulating angiogenesis. However, there are instances in which pro-tumor and anti-tumor properties are present for M1 and M2 macrophages, respectively. M1 macrophages due to the consistent tissue damage in chronic inflammation are linked to enhanced tumor development and progression [49,50]. Additionally, even though TGF-β signaling serves as a promoter of tumor growth and metastasis by boosting tumor cell invasiveness and migration and promoting chemo-resistance in the late stages of cancer, in the early stages, it tends to suppress tumorigenesis [51,52]. Notably, macrophages can polarize and adapt to different functional states, based on the signals and input that they receive from their microenvironment [53,54]. At the same time, they can trans-differentiate, presenting a whole continuum between these two terminal states, M1 and M2 [5].

In conclusion, our data position OXER1 as a prominent tumor-inflammatory component, promoting both macrophage migration toward the tumor mass and tumor cell migration through the TME, possibly facilitated by the M2 macrophage-secreted MMPs [55]. In this respect, specific OXER1 antagonists modifying actin cytoskeleton and cell migration [11,29] are prominent candidates as novel anti-cancer agents.

## 4. Materials and Methods

### 4.1. Cell Cultures and Materials

THP-1 cells were kindly provided by Prof. Papakonstanti (School of Medicine, University of Crete, Heraklion, Greece). The DU-145 and T47D cell lines were purchased from DSMZ (Braunschweig, Germany). These cell lines and THP-1 differentiated macrophages were cultured in RPM1 medium supplemented with 10% FBS and 1% penicillin–streptomycin. White blood cells (WBCs) were cultured in DMEM, 4.5 g/L glucose medium supplemented with 10% FBS and 1% penicillin–streptomycin. Both primary cells and cell lines were cultured at 37 °C, 5% CO_2_ in a humidified tissue culture incubator.

Unless otherwise stated, all media were purchased from Fisher Scientific (part of Thermo Fisher Scientific, Waltham, MA, USA), and all chemicals from Sigma (St. Louis, MO, USA). Cells were treated either with testosterone–BSA (testosterone carboxymethyl-oxime–bovine serum albumin) (10^−6^ M) (T3392), dihydrotestosterone (DHT) (10^−6^ M), 5-oxo-ETE (10^−6^ M), (Cayman Chemical Company, Ann Arbor, MI, USA), or Gue 1654 (10^−6^ M) (Tocris Bioscience, Bristol, UK) or their combinations. Appropriate vehicles were used as controls for each treatment.

### 4.2. WBC Isolation

WBCs were isolated from blood donations from healthy volunteers between 18 and 65 years of age—protocol approval A.P. 32_24022020. Whole blood specimens were obtained in heparin tubes and Histopaque-1077 was added (1:1 to blood sample volume) and samples were centrifuged at 800× *g* for 20 min in order to isolate the PBMCs and granulocytes.

After centrifugation, for isolation of PBMCs, the white blood cell ring fraction containing monocytes and lymphocytes was transferred in a new tube. Cells were washed with PBS twice, fresh medium was added (DMEM, 4.5 g/L glucose, 10% FBS, 1% penicillin–streptomycin) and cells were incubated at 37 °C, 5% CO_2_. Two hours later, the non-adherent as well as the adherent cells, containing the lymphocytes and monocytes respectively, were collected and incubated separately for 24 h. Prior to mRNA isolation, cells were stained with the appropriate markers and assayed via FACS in order to determine the population of mononuclear cells. All cells were stained with CD45, a WBC marker, and combinations of CD3, CD4, CD8, CD19, and CD20 were used for lymphocytes and CD14 and CD11b for monocytes (Appendix A). Cell populations were used for subsequent experiments only when they were >80% pure. For granulocyte isolation, after white blood cell ring fraction removal, the Histopaque and plasma layer were also discarded. Ammonium chloride solution (0.8% NH_4_Cl, 0.1 mM EDTA, buffered with KHC03 for final pH 7.2) was added to the remaining pellet containing the red blood cells (RBCs) and granulocytes in order to lyse the RBCs. After 10 min incubation on ice, the samples were centrifuged for 10 min, 500× *g*. The pellet was resuspended with PBS containing 2% FBS and cells were centrifuged at 120× *g* for 10 min. The pellet containing the polymorphonuclear cells was finally resuspended with PBS, 2% FBS. Cells were stained with CD45, CD16, and CD11b (Appendix A) and assayed via FACS prior to mRNA isolation.

### 4.3. THP-1 Differentiation to Macrophage-Like Cells

THP-1 cells were differentiated as previously described [56] into non-activated macrophages (M0) using phorbol 12-myristate 13-acetate (PMA, 20 ng/mL). After 24 h, the PMA treatment was removed and cells were left to rest for 24 h in RPMI medium supplemented with 5% FBS. M0 cells were then polarized in M1 or M2 macrophages via 48 h incubation with IFN-γ (20 ng/mL) and LPS (250 ng/mL) or IL-4 (20 ng/mL), respectively (Appendix A).

THP-1 differentiation and polarization were validated using RT-PCR and flow cytometry (Appendix A). For RT-PCR, total RNA of THP-1-derived macrophages was isolated using a Total RNA Isolation Kit (NucleoSpin RNA II, NucleoSpin RNA L, Macherey-Nagel) following the manufacturer’s instructions and subjected to RT-PCR. cDNA was synthesized using a TaKaRa PrimeScript™ RT Reagent Kit (Perfect Real Time, 6110). A 10 μL mixture of 1μg RNA, 2μL 5X PrimeScript Buffer (for Real Time), 0.5 μL PrimeScript RT Enzyme Mix I, 0.5μL Oligo dT Primer (50 μM), 0.5μL random hexamers (100 μM), and RNase-free dH_2_O was made and incubated at 37 °C for 15 min for each sample. Generated cDNA was used to perform the amplification in a 20 μL reaction buffer, containing 5 μL of cDNA, 10 μL of master mix (KAPA SYBR FAST qPCR Master Mix, Kapa Biosystems, Inc. Wilmington, MA, USA), 3 μL of dH_2_O, 1 μL of the 3′-primer, and 1 μL of the 5′-primer. For amplification of the different macrophages’ CD markers as well as different cytokines that are highly expressed after differentiation in M1 and M2 (Appendix A), PCR analyses were performed in an Applied Biosystems Step-One Plus^®^ apparatus, using the appropriate primer pairs (Appendix A) (synthesized by Eurofins Genomics, Ebersberg, Germany).

For flow cytometry (see details below), cells (10^6^ cells/mL) were stained with the appropriate primary antibodies [CD68 and CD206 (Appendix A) for M1 and M2, respectively].

### 4.4. Conditioned Medium Derived from Differentiated Monocytes

In order to obtain differentiated monocytes’ conditioned media (CM), after THP-1 differentiation, fresh cultured medium was added to M1 and M2 cells. Forty-eight hours later, the medium was collected and centrifuged twice for 8 min at 800 rpm to remove any remaining cell debris and diluted 1:1 with fresh medium.

### 4.5. Immunocytochemistry (ICC)

Immunocytochemistry for OXER1 was performed using the UltraVision LP Detection System: HRP Polymer Quanto (Thermo Scientific, Cheshire, UK) with diaminobenzidine (DAB) as the chromogen for detection and OXER1-specific antibody (Appendix A). Cells were incubated for one hour with the primary antibody at room temperature and counterstaining was performed using Harris’s hematoxylin (BIOSTAIN, Manchester, UK). Slides were observed and photographed using an optical microscope (Olympus BX41) under identical exposure conditions.

### 4.6. Calcium Changes

For the measurement of calcium changes, the protocol used was as previously described [15]. THP-1 and THP-1-differentiated macrophages were collected, washed, and resuspended in Ca^2+^-containing-medium (140 mM NaCl, 5 mM KCl, 1 mM MgCl_2_, 2 mM CaCl_2_, 10 mM HEPES (N-(2-Hydroxyethyl)-piperazine-N′-(2-ethanesulfonic acid)), 5 mM D-glucose, pH 7.4), at a cell density of 10^6^ cells/mL. Afterwards, they were incubated for 45 min at 37 °C in the dark, with 5 μM fluo-4 AM (Abcam, AB241082), centrifuged at 1500 rpm for 10 min, and resuspended in calcium-free media (without calcium ions, plus 1 mM EGTA, final concentration of calcium ions < 0.1 nM). The cell samples were then transferred to quartz cuvettes and 5-oxo-ETE, testosterone–BSA (testo–BSA), Gue 1654, dihydrotestosterone (DHT), cyproterone acetate (CPA), or their combinations (final concentration of 10^−6^ M) were added to each sample after a few measurements as a baseline. Cytosolic calcium ion responses were expressed as the peak fluorescence intensities (measured at 509 nm) produced with fluo-4 AM using an excitation wavelength of 488 nm. Fluorescence was measured with a PerkinElmer LS-3B fluorescence spectrometer.

### 4.7. Actin Cytoskeleton Visualization

THP-1 cells were seeded at 5 × 10^4^ cells/well in 8-well chamber slides (Nunc™ Lab-Tek™ II Chamber Slide™ System, Thermo Scientific), differentiated into macrophages, and polarized to M1 macrophages. After the polarization, the cells were treated with 5-oxo-ETE, testosterone –BSA, Gue 1654, DHT, or their combinations (final concentration 10^−6^ M) for 20 min. Similarly, DU-145 and T47D cells were seeded at 5 × 10^4^ cells/well in 8 chambers and treated with CM, in the presence or absence of OXER1 ligands, under the same conditions as THP-1. The cells were then fixed in 4% paraformaldehyde (PFA) for 10 min and permeabilized with 0.5% TritonX100 for 10 min. Subsequently, cells were then incubated with 2% BSA for 15 min, followed by rhodamine-labeled phalloidin staining [57] (in 0.2% BSA) for 45 min. The slides were incubated with DAPI for 10 min and mounted with mounting medium (Inova Diagnostics, Inc., San Diego, CA, USA). Visualization was obtained using a confocal laser scanning module attached to a microscope equipped with an argon–krypton ion laser (CLSM, Leica TCS-NT) and representative images of each condition were acquired. The obtained images were quantitatively analyzed by manually marking each cell (60 cells per condition) with NIH software ImageJ (Fiji 2.9.0) after their conversion to grayscale. For macrophages, the morphological changes in actin cytoskeleton and lamellipodia formation were determined by measuring the cell area. For DU-145 and T47D, box counting in fractal dimension analysis was used for the converted binary images to assess the complexity changes due to actin polymerization leading to stress-fiber formation and changes in focal adhesions. More specifically, for cell area measurements, each cell per field was manually selected (polygon selection) and the selected area (ROI) was measured in pixels (Analyze → Measure). The average area was calculated and statistically analyzed with GraphPad Prism. The results are given in the graphs as average % control [Average (AreaTREATMENT × 100/AreaCONTROL)]. For fractal dimension measurements, grayscale images were converted to binary (Process → Binary → Make Binary). Each cell per field was then manually selected (polygon selection) and fractal box count analysis (Analyze → Tools → Fractal Box Count) was used to automatically measure the fractal dimension (D) of the actin cytoskeleton. The average fractal dimension (D) was calculated and statistically analyzed with GraphPad Prism. The results are given in the graphs as average % control [Average (DTREATMENT × 100/DCONTROL)].

### 4.8. Immunofluorescence (IF) for Flow Cytometry (FACS) or Confocal Microscopy

Cells were collected, fixed with 4% PFA, and permeabilized with 0.1% TritonX100 buffer containing 1 M MgCl_2_ and 0.5% fish skin gelatin for 20 min. Cells were then incubated with primary antibodies (Appendix A) for 1 h, followed by a 40 min incubation with the secondary antibodies (Appendix A). Cells were then resuspended in PBS at a density of 10^5^ cells/mL and either analyzed by flow cytometry (using the Attune Acoustic Focusing Cytometer (Thermo Fischer Scientific, Waltham, MA USA)) at a sample size of 2 × 10^4^ cells gated based on forward and side scatter, or seeded in slides, mounted with mounting medium (Inova Diagnostics, Inc., San Diego) containing DAPI, and observed using an inverted confocal scanning microscope (Leica SP5). Both Attune Acoustic Focusing Cytometer software v1.2.5.3891 and Kaluza Analysis version 2.2.1 software were used for the FACS results analysis.

### 4.9. Migration Assays

For M1 or M2 macrophages’ migration, a transwell system was utilized [58]. Cells were seeded at a density of 15 × 10^4^ in the upper compartment of a transwell system (membrane pore size of 0.8 μm, costar) that contained in the lower compartment 5-oxo-ETE, testosterone–BSA, Gue 1654, their combinations (final concentration 10^−6^ M), or medium without treatment (control), with or without the presence of cancer cells (seeding density 3 × 10^5^). After 48 h incubation, macrophages were washed with PBS, and non-migrated cells were removed from the top of the membrane using cotton-tipped applicators. The cells were fixed for 20 min with 100% methanol and dyed with 0.5% crystal violet for 20 min. Representative photos of different fields of view were taken with a phase-inverted microscope (Vert.A1 ZEISS International, Jena, Germany) and the numbers of migrated cells were counted using ImageJ.

For the wound healing assay, DU-145 and T47D cells were seeded in 12-well plates. At 90% confluency, cells were treated with 10 μg/mL mitomycin C (TORIS, United Kingdom) in serum- and antibiotics-free medium for 2 h. A sterile 200 μL pipette tip was used to scratch (wound) a vertical line in the cell monolayer. Cells were observed under an inverted phase-contrast microscope (Vert.A1 ZEISS International, Jena, Germany) and images were taken for time point = 0 (100× magnification). The cells were then treated with OXER1 ligands or macrophages’ CM and the wounded area was photographed after 24 and 48 h. The images were analyzed using PowerPoint and GraphPad Prism 8.4.3.

### 4.10. cAMP Assay

THP-1 cells were seeded at an initial density of 2 × 10^4^ cells per well and differentiated in 96-well flat white plates. cAMP generation was stimulated by forskolin (10^−6^ M) and cells were simultaneously incubated with 5-oxo-ETE (10^−7^ M) for 30 min or the appropriate vehicle. Pre-treatment for 15 min with testosterone–BSA (10^−6^ M) was applied for some of the samples. cAMP generation was assayed using a gain-of-signal competitive immunoassay (Promega V1501 Camp-Glo^TM^ Assay, Promega Corporation, Madison, WI, USA) according to the manufacturer’s instructions. The produced luminescence signal was read in a Tecan M200 Infinite Pro Microplate Reader.

### 4.11. Statistical Analysis

Analysis was performed with GraphPad Prism 8.0.1 (GraphPad Software Inc., San Diego, CA, USA). For each analysis, the homogeneity of variance and normality of data were determined. If the assumptions of normality or homogeneity of variance were not met, a nonparametric statistical test was used, otherwise we used a parametric statistical test for the analysis. *p* < 0.05 was retained as a significance threshold.

## Figures and Tables

**Figure 1 molecules-29-00224-f001:**
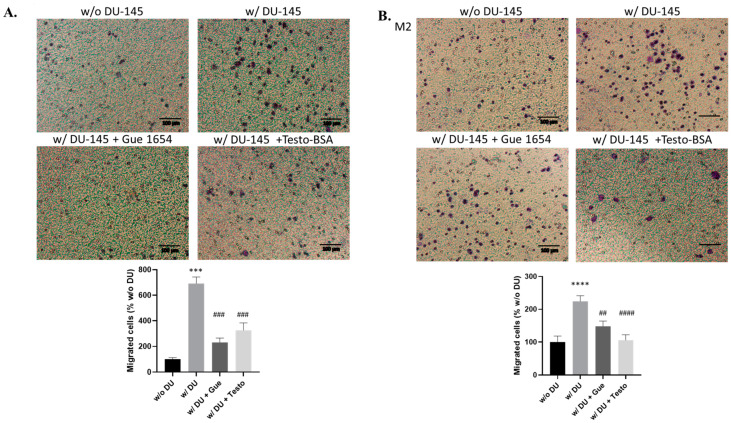
Co-culture of macrophages with DU-145 prostate cancer cells using transwell inserts. Representative photos of M1 (**A**) and M2 (**B**) cell migration after 48 h treatment and staining with crystal violet are shown. M1 and M2 macrophages were seeded on the upper compartment of a transwell insert and media (w/o DU) or DU-145 treated with testosterone-BSA (10^−6^ M), Gue 1654 (10^−6^ M), or the appropriate vehicle were placed on the lower compartment. Migrated cells were counted using ImageJ. Results are shown as mean ± SE of three different experiments. A *t*-test was used to determine the significance between control dataset and treatment dataset. Statistical significance vs. w/o DU *** *p* < 0.001, **** *p* < 0.0001, vs. w/DU ## *p* < 0.01, ### *p* < 0.001, #### *p* < 0.001.

**Figure 2 molecules-29-00224-f002:**
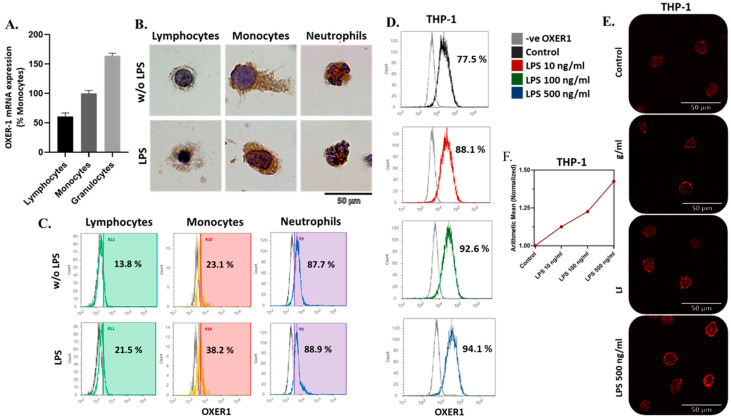
OXER1 mRNA (**A**) and protein expression levels in human WBCs (n = 10) (**B**,**C**) and THP-1 monocytes (n = 3) (**D**,**E**) with or without LPS treatment. WBCs were treated with 1μg/mL LPS and THP-1 with different LPS doses (10, 100, or 500 ng/mL) for 12 h. OXER1 gene expression was assayed using qRT-PCR and protein expression was assayed via ICC (images shown have been selected from mixed leukocyte preparations) (**B**), FACS (**C**,**D**), or IF (**E**). Results are shown as mean ± SE of three independent experiments for mRNA expression (**A**) or representative photos and FACS graphs of three independent experiments are shown (gray histogram in (**C**,**D**) is OXER1-negative). FACS results of LPS dose response in THP-1 cells (**D**) are also given as the normalized arithmetic mean (**F**).

**Figure 3 molecules-29-00224-f003:**
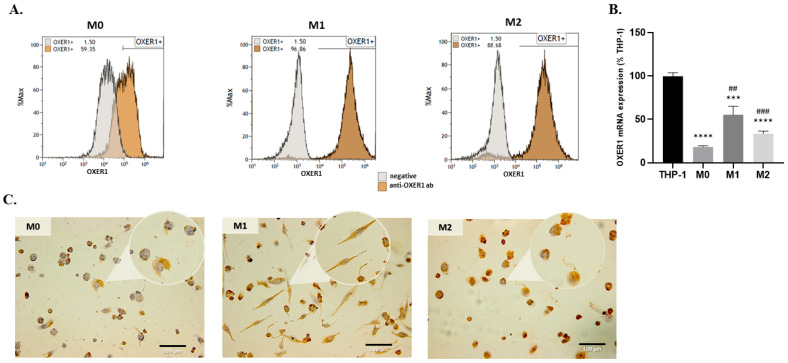
OXER1 mRNA (**B**) and protein (**A**,**C**) expression levels in THP-1, non-activated (M0), and activated (M1, M2) macrophages. OXER1 gene expression was assayed using qRT-PCR and protein expression was assayed via FACS (**A**) and ICC (**C**). Each experiment was assayed in triplicate with appropriate negative controls. Representative photos and FACS graphs of three independent experiments are shown. OXER1 mRNA expression results are shown as mean ± SE. A *t*-test was used to determine the significance between THP-1 or the M0 dataset and different macrophage datasets. Statistical significance vs. THP-1 *** *p* < 0.001, **** *p* < 0.0001, vs. M0 ## *p* < 0.01, ### *p* < 0.001.

**Figure 4 molecules-29-00224-f004:**
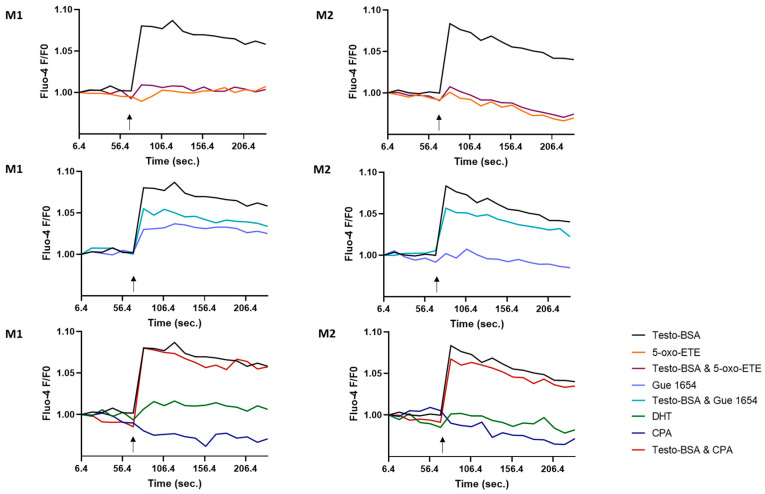
Effect of OXER1 (testosterone–BSA, 5-oxo-ETE, Gue 1654, DHT) and AR ligands (DHT, CPA) on intracellular calcium levels in M1 and M2 cells. Calcium measurements were performed with fluo-4 and all agents were added at a final concentration of 10^−6^ M. Arrows indicate time of agent addition. Each experiment was assayed in triplicate.

**Figure 5 molecules-29-00224-f005:**
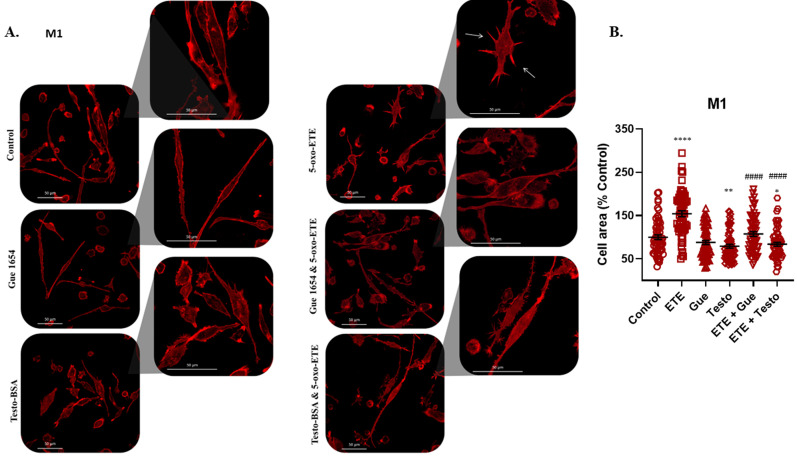
Redistribution of actin filaments in M1 macrophages by 5-oxo-ETE (10^−6^ M), testosterone–BSA (10^−6^ M), Gue 1654 (10^−6^ M), their combination, or the appropriate vehicle (control). Cells were treated for 30 min and filamentous actin redistribution was assessed using rhodamine–phalloidin. Representative photos of three independent experiments are presented (**A**). White arrows indicate the formation of lamellipodia on the cytoplasmic membrane. (**B**) Lamellipodia formation in macrophages was determined by measuring the cell area of 60 cells per treatment. Results are shown as mean ± SE. Each red shape represents a cell. A *t*-test was used to determine the significance between the control, or the 5-oxo-ETE dataset, and the treatment dataset. Statistical significance vs. control * *p* < 0.05, ** *p* < 0.01, **** *p* < 0.0001, vs. 5-oxo-ETE #### *p* < 0.0001.

**Figure 6 molecules-29-00224-f006:**
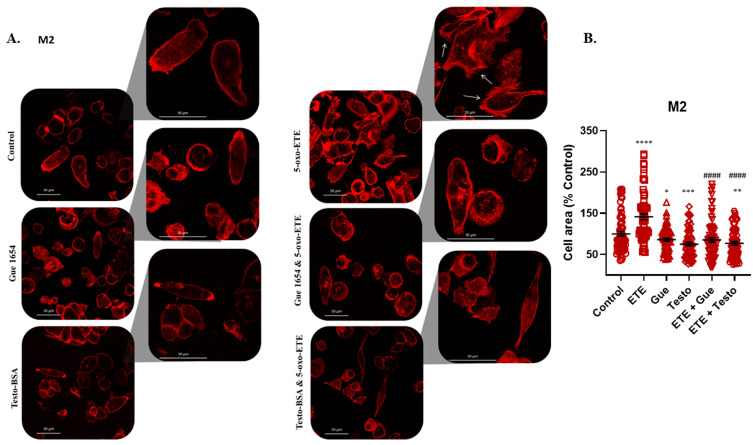
Redistribution of actin filaments in M2 macrophages by 5-oxo-ETE (10^−6^ M), testosterone–BSA (10^−6^ M), Gue 1654 (10^−6^ M), their combination, or the appropriate vehicle (control). Cells were treated for 30 min and filamentous actin redistribution was assessed using rhodamine–phalloidin. Representative photos of three independent experiments are presented (**A**). White arrows indicate the formation of lamellipodia on the cytoplasmic membrane. (**B**) Lamellipodia formation in macrophages was determined by measuring the cell area of 60 cells per treatment. Results are shown as mean ± SE. Each red shape represents a cell. A *t*-test was used to determine the significance between the control, or the 5-oxo-ETE dataset, and the treatment dataset. Statistical significance vs. control * *p* < 0.05, ** *p* < 0.01, *** *p* < 0.001, **** *p* < 0.0001, vs. 5-oxo-ETE #### *p* < 0.0001.

**Figure 7 molecules-29-00224-f007:**
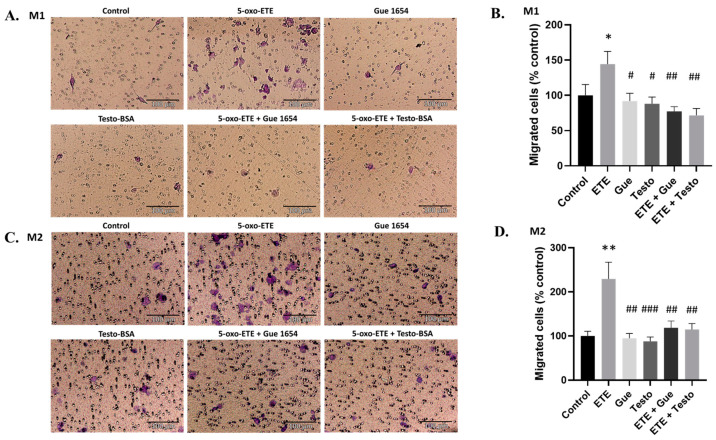
OXER1 ligands’ effect on the migration of M1 (**A**,**B**) and M2 (**C**,**D**) macrophages. Representative photos of three independent experiments for M1 (**A**) and M2 (**C**) cell migration after 48 h treatment and staining with crystal violet are shown. Cells were seeded on the upper compartment of a transwell insert and treatments of 5-oxo-ETE (10^−6^ M), testosterone–BSA (10^−6^ M), Gue 1654 (10^−6^ M), their combination, or the appropriate vehicle (control) were placed on the lower compartment. (B and D) Migrated cells were counted using ImageJ. Results are shown as mean ± SE of three different experiments. A *t*-test was used to determine the significance between the control or the 5-oxo-ETE dataset, and the treatment dataset. Statistical significance vs. control * *p* < 0.05, ** *p* < 0.01, vs. 5-oxo-ETE # *p* < 0.05, ## *p* < 0.01, ### *p* < 0.001.

**Figure 8 molecules-29-00224-f008:**
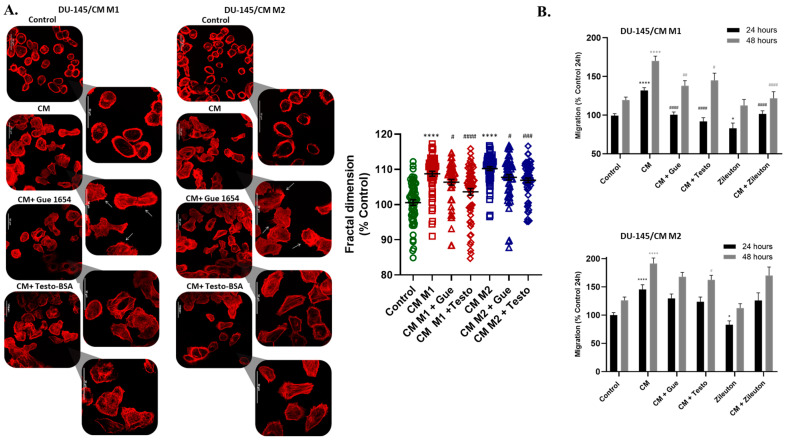
Redistribution of actin filaments (**A**) and changes in the migration of DU-145 (**B**) after treatment with conditioned medium (CM) from M1 or M2 macrophages. DU-145 cells were treated with CM in the presence or absence of testosterone–BSA, Gue 1654, and zileuton. (**A**) Cells were treated for 30 min and filamentous actin redistribution was assessed using rhodamine–phalloidin. Representative photos of three independent experiments are shown. White arrows indicate the formation of filopodia on the cytoplasmic membrane of the cancer cells. Fractal dimension analysis was used to determine the complexity changes due to actin polymerization of 60 cells per treatment. Results are shown as mean ± SE. A *t*-test was used to determine the significance between the control, or the CM dataset, and the treatment dataset. Statistical significance vs. control **** *p* < 0.0001, vs. CM # *p* < 0.05, ### *p* < 0.001, #### *p* < 0.0001. (**B**) Migration of DU-145 after 24 and 48 h. Results are shown as mean ± SE of three independent experiments. A *t*-test was used to determine the significance between the control or the CM dataset, and the treatment dataset. Statistical significance vs. control * *p* < 0.05, **** *p* < 0.0001, vs. CM # *p* < 0.05, ## *p* < 0.01, #### *p* < 0.0001.

## Data Availability

Data are contained within the article and Appendix A.

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
