# Peer review of "5-Oxo-ETE/OXER1: A Link between Tumor Cells and Macrophages Leading to Regulation of Migration"

_molecules, 2023, doi:10.3390/molecules29010224_

Round 1

Reviewer 1 Report

Comments and Suggestions for Authors

Comments on the Quality of English Language

1.       “Reverted” is used incorrectly many times in the manuscript. In most cases it should be replaced by “reversed”, “blocked”, or “inhibited”.

2.       Page 5, line 130: dictated → detected

Reviewer 2 Report

Comments and Suggestions for Authors

Kalyvianaki et al. manuscript is interesting for some reasons. Firstly, these data aimed to strengthen the link between tumor cells and macrophages. Moreover, these results suggest the interplay of epithelial cancer cells and macrophages through OXER1 receptor that is implicated in the migratory/metastatic capacity of tumors thus highlighting that OXER1 antagonists could represent prominent candidates as novel anti-cancer agents. However, there are some critical points that authors are invited to consider.

-      Authors are invited to provide a complete revision of the manuscript to correct syntax and control that all abbreviations are reported.

-      The abstract should be revised to provide a more relevant summary.

-      In the Introduction section, authors write:”…OXER1 binds also testosterone [11]”. To provide a more complete background, authors should provide other references.

-      In the Section 2.4 authors present cytokines mRNA expression. However, authors are invited to also evaluate their release before drawing any firm conclusions, as declare in lines 145-147.

-      In material and method section appropriate citations could be add to better support methods used. Authors should also prepare a paragraph on statistical analysis used.

-      Authors should include in all the figure legends the test used for the statistical analysis.

-      In the Discussion Section, authors are invited to indicate literature data on breast cancer cells that can further support their data on OXER1 antagonists in migration.

Reviewer 3 Report

Comments and Suggestions for Authors

Konstantina Kalyvianaki et al. reviewed the effect of OXER1 in macrophages, and its role in the interaction between macrophages and cancer cells. They found that 5-oxo-ETE, through OXER1, can attract macrophages to the tumor site, and 5-oxo-ETE can enhance cancer cell migration, when which is secreted from macrophages. I believe the results are of interest. However, there are several suggestions need to be addressed before publication.

Major revisions:

1. In the abstract, the background was too long from line 8 to line 16. The abstract should mainly introduce the method and the results, please revise it.   

Minor comments:

1. In line 13, OXER1 was first appeared; please provide its full name.

2. In line 78, M2 (C) should be replaced by M2 (B).

3. In Figure 2, image resolution of all figures was too low to difficult to identify the results, please change them.

4. In line 161, Ca2+ should be replaced by Ca2+.

5. Why did the results of cytokines IL-1b, TNF-α, IL-10 and IL-12 supply in the supplemental files?

Comments on the Quality of English Language

 Minor editing of English language required
